# High Levels of Tumor miR-187-3p—A Potential Tumor-Suppressor microRNA—Are Correlated with Poor Prognosis in Colorectal Cancer

**DOI:** 10.3390/cells11152421

**Published:** 2022-08-05

**Authors:** Lui Ng, Timothy Ming-Hun Wan, Deepak Narayanan Iyer, Zheng Huang, Ryan Wai-Yan Sin, Abraham Tak-Ka Man, Xue Li, Dominic Chi-Chung Foo, Oswens Siu-Hung Lo, Wai-Lun Law

**Affiliations:** Department of Surgery, Li Ka Shing Faculty of Medicine, The University of Hong Kong, Hong Kong SAR, China

**Keywords:** miR-187-3p, colorectal cancer, tumor suppressor, platelet, neutrophil

## Abstract

Background: The microRNA miR-187-3p plays antitumor roles in a variety of cancers. We and others have previously identified miR-187-3p as a potential tumor suppressor in colorectal cancer (CRC), but there are also reports revealing that high miR-187-3p levels are associated with poor prognosis among CRC patients. This study further investigated the clinicopathological significance of miR-187-3p in CRC. Methods: MiR-187-3p levels in paired polyp/CRC/normal specimens or primary CRC/liver metastasis specimens were determined by qPCR, and correlated with the patient’s clinicopathological and postoperative survival data. The clinical findings were validated using our validation cohort and data obtained from the TCGA or GEO databases. The functional effects of miR-187-3p were investigated through its overexpression in CRC cell lines. Results: MiR-187-3p was significantly repressed in colorectal polyps and CRC when compared to adjacent normal tissue. Overexpression of miR-187-3p in CRC cell lines impaired colony formation, cell migration, and invasion, and induced chemosensitivity. Clinical analysis revealed that despite miR-187-3p being repressed in CRC, high tumor miR-187-3p levels were positively correlated with tumor stage and disease recurrence. Further analysis showed that miR-187-3p levels were lower in metastatic specimens when compared to paired primary CRC, suggesting that high tumor miR-187-3p levels resulted from the dissemination of metastatic tumor cells. Tumor miR-187-3p levels were positively correlated with peripheral inflammation-related blood markers. Finally, SPRY1 was identified as a novel target gene of miR-187-3p, and was involved in miR-187-3p-impaired CRC metastasis. Conclusions: This study demonstrated that in spite of its repression and role as a tumor suppressor in CRC, high levels of miR-187-3p in tumors were correlated with poor prognosis and higher levels of peripheral inflammation-related blood markers.

## 1. Introduction

Colorectal cancer (CRC) is the second leading cause of cancer-related deaths worldwide, and is responsible for 10.2% of new cases overall [1]. Despite the increasing number of studies on improving the detection and treatment of CRC, the overall poor prognosis of patients with advanced CRC remains an unresolved predicament [2,3]. Therefore, an in-depth understanding of the molecular networks regulating the development, progression, and therapeutic response of CRC is crucial for identifying effective therapeutic avenues to combat CRC. The investigation of microRNAs (miRNAs) enables a more comprehensive insight into the mechanisms of CRC’s development and progression. MiRNAs are 19–23 nucleotides in length, highly conserved, and code for 1–3% of the human genome. Over 50% of genes are predicted to contain 3′-untranslated regions (3′-UTRs), where miRNAs can bind and regulate their expression. Depending on the degree of complementarity, miRNA binding may cause either mRNA translational repression or, with full complementarity, result in degradation [4]. Considering that such small non-coding RNAs have a significant influence on the overall process of CRC’s occurrence and tumorigenesis, they present some potential therapeutic strategies for CRC. Although there are no miRNA-targeting agents specific to CRC that are available for clinical use at present, several miRNA-targeting drugs/strategies for CRC are currently under research, and have presented remarkable promise in vitro and in vivo [5]. Hence, with a deeper understanding of the mechanism of action of miRNAs and their regulation, improved treatment approaches can be designed to specifically target these miRNAs.

MiR-187-3p, a cancer-related miRNA, was discovered to be repressed and serve as a tumor suppressor in a majority of cancer types, including hepatocellular carcinoma [6], gastric cancer [7], lung cancer [8], renal cell carcinoma [9], osteosarcoma [10], cervical cancer [11], and prostate cancer [12]. However, the upregulation of miR-187-3p was also found in gastric cancer [13], oral carcinoma [14], ovarian cancer [15], and breast cancer [16], where it was involved in supporting tumor progression. These findings suggest heterogeneous roles of miR-187-3p in promoting or suppressing carcinogenesis among different cancer types. We previously identified the dysregulated expression of 39 miRNAs during the transformation process of colorectal cells, with miR-187-3p being one of the most repressed miRNAs in both colorectal polyp and CRC in comparison with adjacent normal tissue, suggesting its repression to be an early event in colorectal cells’ transformation [17]. Indeed, miR-187-3p has been shown to be downregulated in CRC, and to functionally regulate tumor cells’ proliferative and invasive behavior. In addition, low miR-187 levels were associated with worse overall survival rate and/or relapse-free survival rate [18,19]. On the other hand, a more recent article reported that high miR-187-3p levels were correlated with poor prognosis among CRC patients, and miR-187-3p was one of the microRNAs within the five-microRNA signature for predicting prognosis [20]. Further investigation is still required to better understand the clinical significance of miR-187-3p in CRC and other cancers.

This study further investigated the functional effects and clinical significance of miR-187-3p in CRC.

## 2. Materials and Methods

### 2.1. Patient Recruitment

The CRC patients who underwent surgical tumor resection at the Department of Surgery, Queen Mary Hospital, Hong Kong, were recruited after providing their written consent between the years 2007 and 2013. The validation cohort consisted of tumor and adjacent normal tissue specimens collected from CRC patients from 2012 to 2017. All enrolled patients received no perioperative chemotherapy. Tissue specimens were snap-frozen in liquid nitrogen and stored at −80 °C until further molecular analysis. Clinical and pathological data were retrieved from the patients’ medical records, including gender, age, diagnosis, location and size of tumor, TNM staging, occurrence of distant metastases, local invasion, and differentiation. The study was approved by the Institutional Review Board of the University of Hong Kong/Hospital Authority Hong Kong West Cluster (HKU/HA HKW IRB), and written informed consent was obtained from the patients prior to their inclusion.

### 2.2. Cell Culture

The human CRC cell lines HCT116, HT29, and SW480, along with the 293TN producer cell line, were obtained from the ATCC and maintained in high-glucose Dulbecco’s modified Eagle’s medium (DMEM) supplemented with 10% fetal bovine serum, along with 100 U/mL and 100 µg/mL of penicillin and streptomycin, respectively, at 37 °C in a humidified incubator supplemented with 5% CO_2_.

### 2.3. Transfection

Transient transfections were performed using Lipofectamine 3000 (Invitrogen, Carlsbad, CA, USA), according to the manufacturer’s instructions. Briefly, 90 pmol of mirVana hsa-mir-187-3p mimic, negative control, or inhibitor (Ambion, Austin, TX, USA) was used. For DNA transfections, the procedure was similar, except 2500 ng of DNA was used and mixed with 5 µL of P3000 reagent.

### 2.4. Construction of miR-187-3p Expression Plasmids

In order to synthesize the miR-187-3p and SPRY1 inserts, reverse transcription (RT) was first performed with Superscript III reverse transcription reagent (Invitrogen), which was then used to amplify the miR-187-3p and SPRY1 inserts via polymerase chain reaction using Platinum Taq DNA Polymerase High Fidelity (Invitrogen) and the following primers: miR-187-3p EcoR1 forward primer (5′-CCGGAATTCCACCTGGAGCACAGGTCATC) and miR-187-3p-BamH1 reverse primer (5′-CGCGGATCCAGCTGTGTACGGAGAGACGA); or SPRY1-Not1 forward primer (5′-CAAAAAGCGGCCGCATGGATCCCCAAAATCAACATG) and SPRY1-Kpn1 reverse primer (5′-GGGAAAGGTACCTCATGATGGTTTACCCTGACC). The PCR product was purified, digested with restriction enzymes, and cloned into the lentiviral vector pCDH-CMV-MCS-EF1-copGFP (Biosciences, Cambridge, UK). The plasmids were transformed into Stbl2-competent cells (Life Technologies, Carlsbad, CA, USA), and the sequence of positive clones was confirmed by DNA sequencing.

### 2.5. Lentiviral Infection

To make the lentivirus for constructing stable miR-187-3p-overexpressing CRC cells, 293TN cells were co-transfected with pPACK plasmids and the miR-187-3p pCDH construct using Lipofectamine 3000 reagent. Then, 1 × 10^6^ 293TN cells were seeded into each 6-well plate, after which 25 µL of pPACKH1 and 2.5 µg of pCMV-187 or pCMV-copGFP alone as a control, with 5 µL of p3000 reagent, were added to 125 µL of serum-free DMEM. Next, 7.5 µL of Lipofectamine 3000 was added to another 125 µL of serum-free DMEM, and both mixtures were incubated for 5 min at room temperature separately, before being mixed and incubated for 15 min at room temperature. This mixture was added dropwise to the 70–90% confluent 293TN cells. After 48 h of transfection, the medium was collected and centrifuged at 3000× *g* for 15 min, after which the viral supernatant was collected and supplemented with 8 µg/mL of polybrene for infection. Finally, 1 × 10^6^ CRC cells were seeded in 6-well plates, and 2 mL of viral supernatant was added and incubated for 48 h to allow infection.

### 2.6. Total RNA Extraction

Total RNA was extracted using the mirVana miRNA isolation kit (Ambion, Austin, TX, USA), according to the manufacturer’s instructions. The RNA concentration was measured using a NanoDrop ND-1000 spectrophotometer (NanoDrop Technologies, Wilmington, DE, USA), and RNA was stored at −80 °C before use.

### 2.7. Quantitative RT-PCR

A DNA-primer-based reverse transcription PCR (RT-PCR) method was applied to determine the miR-187-3p levels, as previously described [17]. For RT-PCR, the total RNA was polyadenylated (New England Biolabs, Ipswich, MA, USA), and was subsequently reverse-transcribed using the PrimeScript RT Reagent Kit (Takara, Shiga, Japan) with a custom oligo dT primer. Quantitative estimation of targets was performed using a ViiA 7™ Real-Time PCR System (Applied Biosystems, Waltham, MA, USA) with TB Green Premix Ex Taq II detection chemistry (Takara, Shiga, Japan). The expression of miR-187-3p was normalized by U6, and miR-187-3p expression in normal or tumor cells was expressed as 2^−^^∆Ct^^(miR−187−3p-U6)^, whereas the fold change of T vs. N was expressed as 2^−^^∆∆Ct^^(T−N)^. Experiments were performed in duplicate.

### 2.8. Wound-Healing Assays

Stably transfected CRC cell lines with pCMV-187 or an empty vector were seeded at 1 × 10^5^ cells in a 24-well plate and grown to 100% confluence. A p200 pipette tip was used to create a wound, and the cells were washed with medium to remove excess cell debris. The speed of wound closure was measured and monitored under a light microscope every 24 h until full closure.

### 2.9. Transwell Cell Migration Assay

The migratory ability was measured using the Transwell model (Corning, New York, NY, USA) by seeding 7.5 × 10^4^ cells into the upper chamber of each insert with 250 µL of 1% FBS DMEM, and 750 µL of 10% FBS DMEM was added to the lower part of the chamber as the chemoattractant. After 48 h of incubation at 37 °C in a humidified incubator containing 5% CO_2_, the upper chamber was removed, and unmigrated cells inside the chamber were removed using a cotton swab. Then, the chamber was fixed using 100% cold methanol on ice for 2 min, stained with 0.2% crystal violet solution for 20 min, and then washed with water. The numbers of migrated cells were counted in five random visual fields using an inverted light microscope.

### 2.10. MTT Cell Proliferation Assay

A total of 7500 cells was seeded into a 96-well plate and incubated in DMEM containing 10% FBS at 37 °C in 5% CO_2_. After incubation for the indicated time interval, the medium was removed, and 3-(4,5-dimethylthiazol-2-yl)-2,5-diphenyltetrazolium bromide (MTT) was added at a 20:1 ratio to serum-free DMEM and incubated at 37 °C for 2 h. The medium was then removed, and 100 µL of DMSO was added to lyse the cells and dissolve the formazan crystals. The absorbance was determined at 570 nm using a Multiskan GO Microplate Spectrophotometer (Thermo Fisher, Waltham, MA, USA).

### 2.11. Chemotherapeutic Drug Treatment

After transient transfection or stable transfection, 7500 cells were seeded into a 96-well plate. Four treatment groups were set up: a control group where DMSO was added at a 5 µL/mL concentration, an OXA group where the cells were exposed to 20 µg/mL of oxaliplatin (OXA), a 5-FU group where 100 µg/mL of 5-fluorouracil (5-FU) was added and, finally, a 5-FU + OXA group, where a combination of 100 µg/mL 5-FU and 20 µg/mL OXA was added. The cells were incubated for 48 h in 5% CO_2_ at 37 °C.

### 2.12. Clonogenic Assay

Stable clones of miR-187-3p or the control vector were seeded at a cell density of 100 cells in a 6-well plate. Cells were incubated at 37 °C in 5% CO_2_ for 14–21 days to allow colony formation. The medium was removed, and then the colonies were fixed with 100% ice-cold methanol and stained with 0.2% crystal violet for 20 min at room temperature, followed by washing in water before being air-dried.

### 2.13. MiR-187-3p Target Gene Prediction

Potential gene targets of miR-187-3p were identified using the TargetScanHuman (version 7.2,) program (http://www.targetscan.org) (Cambridge, MA, USA) and miRmap (https://mirmap.ezlab.org/, Lausanne, Switzerland) (accessed on 4 September 2019).

### 2.14. Luciferase Reporter Assay

A potential binding site of miR-187-3p was predicted at position 1148–1154 (5′-GACACG-3′) of SPRY1′s 3′-UTR using the miRmap database (https://mirmap.ezlab.org/, Lausanne, Switzerland)(accessed on 4 October, 2019). Subsequently, a 25 bp oligo duplex containing wild-type SPRY1 3′-UTR (5′-GACACG-3′) or mutant SPRY1 3′-UTR (5′-CAGAGC-3′) was ligated into the pMIR-REPORT luciferase vector (Thermo Fisher Scientific, Waltham, MA, USA) using the MluI and SpeI restriction sites. For the luciferase reporter assay, HCT116 cells (100,000 cells/well) were co-transfected with the wild-type or the mutant reporter vector with miR-187-3p mimic or negative control and Renilla luciferase control vectors, using Lipofectamine 3000 (Invitrogen, Carlsbad, CA, USA). After 48 h, the cells were lysed and processed using the Dual-Luciferase Reporter Assay System (Promega) according to the manufacturer’s instructions. The firefly and Renilla luciferase activity was measured in a CLARIOstar multimode microplate reader (BMG Labtech, Ortenberg, Germany), and the ratio was normalized. Experiments were performed in triplicate.

### 2.15. Statistical Analyses

Data were expressed as the means ± SD of at least three independent experiments. Student’s *t*-test or Fisher’s exact test was used to analyze the differences between 2 groups, and one-way ANOVA was used to analyze the difference among multiple groups. Pearson’s or Spearman’s correlation was applied to determine the presence of correlation between miR-187-3p levels and other parameters. Survival curves were generated using the Kaplan–Meier test and compared by the log-rank test. *p*-Values of less than 0.05 were considered statistically significant. All statistical analyses and graphical representations were performed using SigmaPlot 10.0 software (Systat Software, Erkrath, Germany).

## 3. Results

### 3.1. MiR-187-3p Was Downregulated in Colorectal Polyp and CRC Tissues

To validate our previous miRNA microarray results, which demonstrated that miR-187-3p was repressed in colorectal polyp and CRC tissues [17], DNA-primer-based reverse transcription PCR (RT-PCR) was performed to determine the expression of miR-187-3p in synchronous polyp, CRC, and normal mucosa samples from 14 patients. As shown in Figure 1, miR-187-3p was significantly repressed in polyp (*p* = 0.017) and CRC (*p* = 0.009) tissues when compared to adjacent normal tissue, whereas no significant difference was observed between polyp and CRC tissues. These results indicate that miR-187-3p was indeed repressed during the early transformation process of colorectal cells.

### 3.2. MiR-187-3p Overexpression Inhibited the Colony-Formation Ability of CRC Cells

To investigate the functional effect of miR-187-3p in CRC, we generated stable clones overexpressing miR-187-3p or the control vector from two CRC cell lines: HCT116 and SW480. The overexpression of miR-187-3p in stable clones was confirmed by quantitative RT-PCR (Figure 2A). The effect of miR-187-3p overexpression on the clonogenic ability of CRC cells was investigated (Figure 2B). Stable overexpression of miR-187-3p in HCT116 cells significantly decreased colony formation compared to the empty control vector. The mean number of HCT116-miR-187-3p cells that formed colonies was 36.5, which was significantly lower than the 81.0 colonies formed by HCT116-control vector cells (*p* = 0.0391). Similar findings were observed for SW480 cells. An average of 42.3 colonies was formed from stable SW480-miR-187-3p cells, which was significantly lower than the number formed from SW480-control vector cells (63.0 colonies, *p* = 0.027). These results demonstrate that miR-187-3p overexpression impaired the colony-formation ability of CRC cells.

### 3.3. MiR-187-3p Overexpression Impaired CRC Cell Migration and Invasion

The effect of miR-187-3p overexpression on cell migration was determined by wound-healing assays (Figure 2C). HCT116-miR-187-3p clones migrated at an average speed of 191.8 μm/day, whereas the control vector migrated at an average speed of 277.0 μm/day, indicating an average of 30.8% slower cell migration following miR-187-3p upregulation (*p* = 0.0273). Similarly, stable SW480-miR-187-3p cells also showed significantly slower cell migration compared to the control vector. The SW480 CMV-187 cells migrated at an average speed of 192.3 μm/day, which was significantly slower than that of the control vector cells, which migrated at 224.0 μm/day (*p* = 0.0117).

Migration assays were further performed to validate the effect of miR-187-3p overexpression on CRC cell migration. As shown in Figure 2D, the mean number of cells that migrated for HCT116-miR-187-3p cells was 143.3, which was significantly lower when compared to the control vector (177, *p* = 0.0399). The inhibitory effect of miR-187-3p overexpression on migration was more obvious for SW480 cells. The mean number of stable SW480-miR-187-3p cells that migrated was 97.2, which was significantly lower than the average number of control vector cells that migrated (196.8, *p* = 0.0352).

We also examined the effect of miR-187-3p overexpression on the invasive ability of CRC cells (Figure 2E). Overexpression of miR-187-3p in HCT116 cells significantly reduced the number of cells that invaded compared to the control vector (*p* = 0.0273). Similarly, the number of stable SW480-miR-187-3p cells that invaded was significantly reduced when compared to the negative control (*p* = 0.0485).

### 3.4. MiR-187-3p Overexpression Induced Chemosensitivity of CRC Cells

We next investigated the functional effects of miR-187-3p overexpression on the cellular response to conventional chemotherapeutic drugs used to treat CRC, including 5-fluorouracil (5-FU), oxaliplatin (OXA), and combined treatment (5-FU + OXA). MTT assay was performed to determine the relative residual cell number, in terms of optical density at 570 nm, after 72 h drug or vehicle (DMSO) treatment.

As demonstrated in Figure 3A, stable SW480-miR-187-3p cells and -control vector cells showed similar optical density upon DMSO treatment. On the other hand, the optical density was significantly lower in miR-187-3p cells when compared to control vector cells upon OXA (*p* < 0.01) and combined treatment (*p* < 0.05). The mean optical density for OXA-treated miR-187-3p cells and control vector cells was 0.420 and 0.848, respectively, and 0.410 and 0.785 for combined-drug-treated miR-187-3p cells and control vector cells, respectively, indicating a significant difference of 50.4% and 47.8% survival between stable miR-187-3p-overexpressed and control vector cells upon OXA and combined drug treatment, respectively. A similar trend was observed for 5-FU treatment, but the result was not statistically significant. These results demonstrate that stable miR-187-3p overexpression induced chemosensitivity in SW480 cells.

HCT116-miR-187-3p cells also showed increased chemosensitivity to 5-FU (*p* < 0.05) and OXA (*p* < 0.01) compared to the control vector, as shown in Figure 3B. The mean optical density for miR-187-3p and control vector cells following 5-FU treatment was 0.240 and 0.375, respectively, showing a significant survival difference of 36.0%. The mean optical density for miR-187-3p cells and control vector cells upon OXA treatment was 0.257 and 0.439, respectively, with a significant survival difference of 41.5%. A similar trend was observed for 5-FU + OXA combined treatment, but the result was not statistically significant.

In addition, we investigated whether transient overexpression of miR-187-3p induced drug response in a chemoresistant cell line (DLD1) [21,22]. Transient overexpression of miR-187-3p was confirmed by qRT-PCR (Figure 3C). Figure 3D shows that transient overexpression of miR-187-3p in DLD1 cells yielded a significant difference in response to 5-FU and OXA treatment, as compared to negative-control-transfected cells (*p* < 0.05). The mean optical density for miR-187-3p mimic cells and negative control cells upon 5-FU treatment was 0.354 and 0.604, respectively, indicating a 41.4% difference in survival. The optical density was 0.418 and 0.673 for OXA-treated miR-187-3p and negative-control-transfected cells, respectively, with a 37.9% difference in survival. These results indicate that transient miR-187-3p overexpression significantly induced the response of chemoresistant DLD1 cells to 5-FU and OXA treatment.

### 3.5. Tumor miR-187-3p Levels Were Associated with Age and Stage

To understand the potential roles of miR-187-3p in CRC, we analyzed the correlation between tumor miR-187-3p levels and clinicopathological parameters. The levels of miR-187-3p in CRC and adjacent normal tissues were determined in 54 CRC patients. The miR-187-3p levels were significantly repressed in CRC tissue compared to adjacent normal tissue (Figure 4A, *p* < 0.001). In particular, 36 out of the 54 patients showed at least a twofold decrease in miR-187-3p levels in CRC tissue, indicating that miR-187-3p was indeed significantly repressed during colorectal cell tumorigenesis.

Our clinicopathological correlation analysis showed that tumor miR-187-3p was significantly associated with age. As shown in Figure 4B, the median tumor miR-187-3p level in older patients (above 65 years of age) was 0.00000171, which was significantly lower than that of those below 65 years old (0.00000497, *p* = 0.039).

Since our functional experiments showed that miR-187-3p impaired colony formation, cell migration, and response to chemotherapy, we hypothesized that its level was correlated with the tumor progression and prognosis of CRC patients. We compared the levels of miR-187-3p in CRC of different stages. As shown in Figure 4D, the mean tumor miR-187-3p level was significantly higher for stage II (0.0000197), III (0.0000276), and IV (0.00000630) compared to stage I (0.00000232).

To validate their correlation with age and stage, we determined the miR-187-3p levels in an independent cohort of 160 paired CRC and adjacent normal tissues. MiR-187-3p also demonstrated significant repression in CRC tissue when compared to adjacent normal tissue in the validation cohort (Figure 4D, *p* < 0.0001). Moreover, consistent with the above observation, tumor miR-187-3p levels were associated with patients’ age and tumor stage. The mean tumor miR-187-3p level in patients above 65 years old was 0.00000229, which was significantly lower than that of those below 65 years old (0.00000365, *p* = 0.028). In addition, the mean tumor miR-187-3p levels for stage II (0.000179), III (0.000105), and IV (0.000560) CRC were significantly higher compared to stage I (0.00000393, Figure 4F).

Furthermore, we validated the correlation of miR-187-3p levels with age and tumor stage using TCGA data analyzed at UALCAN on 28 April 2022. As shown in Figure 4G, tumor miR-187-3p levels showed a decreasing trend with ascending age group for both colon and rectal cancer patients, although only the age groups 41–60 vs. 81–100 for colon cancer patients and 21–40 vs. 61–80 for rectal cancer patients were statistically significant. The lower levels of miR-187-3p in older patients might explain why the risk of CRC development increases with age. On the other hand, tumor miR-187-3p levels showed an increasing trend with tumor stage for both colon and rectal cancer patients (Figure 4H). The tumor miR-187-3p levels in stage II–IV tumors were significantly higher or showed a trend of higher levels compared to stage I tumors for both colon and rectal cancer patients.

### 3.6. Metastatic Tumors Showed Lower miR-187-3p Compared to Primary CRC

We hypothesized that the increase in miR-187-3p levels at higher stages of CRC was caused by the escape of CRC cells from the primary tumor, which expressed lower levels of miR-187-3p, i.e., with higher migration ability. To test this hypothesis, we compared the expression levels of miR-187-3p in paired liver metastasis and primary CRC specimens from 10 patients. As shown in Figure 5, miR-187-3p levels were significantly lower in liver metastases when compared to the paired primary CRC specimens (*p* = 0.049). In particular, miR-187-3p levels were lower in liver metastases in 8 out of 10 patients.

We validated this finding using two GEO datasets: GSE98406 (Kelley KA et al. 2017) and GSE56350 (Drusco A and Croce CM 2014). The GSE98406 dataset consists of miRNA profiles from 14 paired FFPE colon tumor and metastasized liver tumor samples derived from the same patient (Figure 5B). The miR-187-3p levels were significantly lower in liver metastases comparing to primary CRC samples (*p* = 0.009). More specifically, 9 out of 14 patients showed lower miR-187-3p levels in liver metastases. The GSE56350 dataset consists of miRNA profiles from 41 colon cancer patients with lymph node metastases, and 15 were diagnosed with colon cancer along with lymph node and liver metastases (any T or N, M1). Figure 5C shows that mir-187-3p levels were significantly lower in lymph node metastasis samples (4.624) compared to primary tumor samples (5.206, *p* = 0.009). There was also a trend of higher miR-187-3p levels (*p* = 0.129) in primary tumor samples (5.852) compared to liver metastases (5.269, Figure 5D) for the 15 paired CRC and liver metastasis samples. More specifically, 10 out of 15 patients showed lower miR-187-3p levels in liver metastases. These results demonstrated that metastasized CRC cells possessed lower miR-187-3p expression compared to primary CRC cells, suggesting that the increase in miR-187-3p levels in higher stages of CRC was due to the escape of low-miR-187-3p-expressing CRC cells from the primary tumor.

### 3.7. High miR-187-3p in CRC Was Correlated with Recurrence

Although miR-187-3p was significantly repressed in polyp and CRC tissues, its levels were increased when the tumor progressed. We hypothesized that tumor miR-187-3p levels were a predictive biomarker for CRC recurrence; hence, we analyzed the correlation between tumor miR-187-3p levels and the development of recurrent disease within 5 years following R0 resection for stage I–III CRC. As shown in Figure 6A, compared to those showing no recurrent disease (*n* = 35), patients who developed postoperative recurrence (*n* = 11) had significantly higher tumor miR-187-3p levels (0.00000822 vs. 0.00000185, *p* = 0.028). Moreover, CRC patients with higher tumor miR-187-3p expression had a significantly worse disease-free survival when compared to those with low tumor miR-187-3p levels (*p* = 0.029, Figure 6B).

### 3.8. Higher miR-187-3p Levels Were Correlated with Elevated Preoperative Peripheral Inflammation-Related Blood Parameters

There has been an increasing number of studies reporting the association between peripheral inflammatory cell markers and relapse-free or overall survival of CRC patients; hence, we further investigated whether tumor miR-187-3p levels were correlated with the immune cell profiles of preoperative blood samples. As shown in Figure 7A, tumor miR-187-3p levels were positively correlated with several peripheral inflammation-related blood cell markers, namely, red cell distribution width (R = 0.404, *p* = 0.0049), platelet count (R = 0.395, *p* = 0.0062), platelet-to-lymphocyte ratio (R = 0.369, *p* = 0.0109), neutrophil count (R = 0.307, *p* = 0.0358), and neutrophil-to-lymphocyte ratio (R = 0.426, *p* = 0.0030). As high red cell distribution width, platelet count, and neutrophil count are all associated with tumor metastasis and poor prognosis, these correlations further confirm that high tumor miR-187-3p levels are associated with inflammatory condition, and are a poor prognostic factor.

We were able to validate these correlations with the validation cohort (*n* = 144). As shown in Figure 7B, tumor miR-187-3p levels showed signification correlations with levels of red cell distribution width (R = 0.267, *p* = 0.0012), platelet count (R = 0.263, *p* = 0.0015), platelet-to-lymphocyte ratio (R = 0.307, *p* < 0.0001), neutrophil count (R = 0.213, *p* = 0.0105), and neutrophil-to-lymphocyte ratio (R = 0.245, *p* = 0.0031).

More specifically, when the CRC patients were stratified into high- and low-miR-187-3p groups, the high-miR-187-3p group had significantly higher red cell distribution width (16.32 vs. 14.84, *p* = 0.005), platelet count (283.24 vs. 239.30, *p* = 0.006), platelet-to-lymphocyte ratio (246.33 vs. 177.19, *p* < 0.001), neutrophil count (5.12 vs. 4.27, *p* = 0.025), and neutrophil-to-lymphocyte ratio (4.49 vs. 3.23, *p* = 0.007) (Figure 8). These findings confirm the close association between tumor miR-187-3p levels and these immune cell parameters.

### 3.9. Identification of SPRY1 as a Direct Target of miR-187-3p

To identify the potential molecular mechanism through which the repressed expression of miR-187-3p impacted the metastatic ability of CRC cells, we used the TargetScan and miRmap databases to identify putative gene targets of miR-187-3p that are commonly associated with carcinogenesis according to the existing literature. Consequently, we identified eight potential gene targets, including protein sprouty homolog 1 (SPRY1), argonaute RISC component 1 (AGO1), centromere protein A (CENPA), torsin family 4 member A (TOR4A), ribonuclease H2 subunit A (RNASEH2A), GIPC PDZ domain-containing family member 1 (GIPC1), leucine-rich repeat and fibronectin type III domain-containing 1 (LRFN1), and neurogenin-2 (NEUROG2).

To further investigate the putative targets of miR-187-3p, we verified the expression of these nine predicted genes within an HCT116 cell line stably transfected with miR-187-3p or the control vector. Fibroblast growth factor 9 (FGF9) was used as a positive control, since it was previously validated as a direct target of miR-187-3p [23]. Quantitative RT-PCR assay showed that only SPRY1 showed a significant downregulation within the miR-187-3p-overexpressing HCT116 cell line compared to the control vector (fold change: −3.12, *p =* 0.0006) (Figure 9A). The rest of the potential gene targets showed no statistically significant changes in expression between the comparison groups.

Subsequently, a dual luciferase reporter assay was performed to test whether SPRY1 functions as a direct target of miR-187-3p. The predicted miRNA binding site within the 3′-UTR regions of SPRY1, as well as a mutated version of the sequence, was used to generate reporter constructs (Figure 9B,C). Later, these reporter constructs were transfected with an miR-187-3p mimic or empty vector, and the luciferase signal was analyzed. Our analysis suggests that within the HCT116 cell line transfected with the original SPRY1 3′-UTR miRNA binding site, the luciferase signal was significantly reduced upon overexpression of miR-187-3p as compared to the control vector (*p =* 0.0133) (Figure 9D). Importantly, there was no statistically significant change in the luciferase signal upon miR-187-3p co-transfection when the 3′-UTR sequence of the SPRY1 reporter plasmid was mutated. These findings indicate that miR-187-3p directly targets SPRY1 through the predicted 3′-UTR binding site.

To test whether miR-187-3p impaired CRC cell migration by repressing SPRY1, cell migration and invasion assays were performed to investigate the effects of SPRY1 overexpression in stable HCT116-miR-187-3p cells or control vector cells. Quantitative PCR was performed to confirm the overexpression of SPRY1 in those cells (Figure 9E). Interestingly, the levels of SPRY1 in the stable HCT116-miR-187-3p cells were significantly lower than those in the control vector cells (*p* < 0.001). As shown in Figure 9F,G, overexpression of SPRY1 significantly induced cell migration and invasion when compared to control vector cells. Moreover, SPRY1 overexpression was able to rescue the impaired cell migration and invasion in stable miR-187-3p cells. These results suggest that miR-187-3p impaired CRC cell migration and invasion, at least partially, through repressing its target gene *SPRY1*.

### 3.10. Correlation of miR-187-3p and SPRY1 Levels in CRC Tissue

To validate the correlation between miR-187-3p and *SPRY1* in clinical specimens, we determined their levels in the 54 paired CRC specimens and adjacent normal mucosa, as mentioned in Section 3.5. *SPRY1* was significantly overexpressed in CRC specimens (Figure 10A, *p* < 0.001). Pearson’s correlation analysis showed that there was a significant inverse correlation between miR-187-3p and *SPRY1* (R = −0.302, *p* = 0.028; Figure 10B), indicating that *SPRY1* was indeed a target gene of miR-187-3p in CRC.

## 4. Discussion

Increasing numbers of studies have been showing that dysregulated miRNA expression can be detected in various cancer types, including CRC, and plays a vital role in tumorigenesis and progression. Identifying the outcomes of such aberrant expression of miRNAs provides the foundations to discover novel strategies for diagnosis and treatment. MiRNA profiling in our pilot experiment demonstrated that miR-187-3p was significantly repressed in both colorectal polyp and CRC tissues when compared to adjacent normal tissue. Hence, this study aimed to unravel the role of miR-187-3p in CRC, along with its potential as therapeutic target.

MiR-187-3p has been found to be downregulated and play a tumor-suppressor role in a majority of cancer types [6,8,9,10,11,12]. Two studies reported the significant downregulation of miR-187-3p and its role in CRC [18,19], suggesting that miR-187-3p is an important negative regulator of CRC progression, as is consistently reported in this and other studies. Furthermore, the results obtained in this study provide additional information on roles of miR-187-3p in CRC, allowing us to understand the functional roles and the underlying mechanism of miR-187-3p in CRC more thoroughly. Our findings are consistent with the results of Zhang et al.’s and Wang et al.’s studies, which found that miR-187-3p inhibition increased the aggressiveness of CRC [18,19]. In addition to targeting CD276 and the SMAD-mediated EMT process as reported in their studies, we further identified SPRY1—which has been reported to induce EMT in CRC [24]—as a novel target gene of miR-187-3p. Overexpression of SPRY1 was able to rescue the impaired cell migration and invasion in stable miR-187-3p cells. These results suggest that miR-187-3p impaired CRC cell migration and invasion, at least partially, through repressing its target gene *SPRY1*.

One notable finding in this study is that although miR-187-3p was repressed in CRC cells, higher levels of miR-187-3p were associated with poor prognosis. Indeed, miR-187-3p demonstrated a similar pattern in hepatocellular carcinoma (HCC)—it is repressed in HCC, and functionally inhibits the metastasis of HCC both in vitro and in vivo [6]. On the other hand, our analysis of TCGA data using YM500v3 showed that high miR-187-3p levels in HCC are associated with poor prognosis (HR: 2.33, *p* = 0.0283, data not shown). This unusual and contradictory observation has also been reported for another miRNA—miR-485-5p, which was repressed in CRC but its high expression was associated with poor prognosis [20]. One common property shared between these miRNAs is that their repression induces cell invasion and migration [25]. Our clinical experiment showed that miR-187-3p levels were higher in stage II, III, and IV CRCs when compared to stage I CRCs; although its levels were significantly lower than the paired non-tumor tissue (data not shown), it is rational to postulate that the association between poor prognosis and high expression of these tumor-suppressor miRNAs is a result of the progression of the tumor and subsequent dissemination of metastatic tumor cells—which express low levels of miR-187-3p or miR-485-5p—from the primary tumor site. Our subsequent experiment, which showed that miR-187-3p levels were significantly lower in liver metastasis specimens compared to paired CRC specimens, supported this postulation.

This study also suggests that tumor miR-187-3p levels are a predictive biomarker for CRC recurrence. Tumor miR-187-3p levels were significantly higher in stage I–III CRC patients who developed recurrent disease within 5 years post-operation, as compared to those who showed no recurrence. Moreover, CRC patients with higher tumor miR-187-3p expression had a significantly worse disease-free survival when compared to those with low tumor miR-187-3p levels. Further investigation in another cohort of patients is warranted to validate these findings. We hope that by identifying a panel of promising biomarkers for predicting the risk of recurrence—such as miR-187-3p in this study—better postoperative treatments can be designed for CRC patients.

Finally, this study demonstrated, for the first time, that tumor miR-187-3p levels are positively correlated with red cell distribution width (RDW), platelet count, platelet-to-lymphocyte ratio (PLR), neutrophil count, and neutrophil-to-lymphocyte ratio (NLR). Although miR-187-3p dysregulation has been reported in various cancers, its association with immune cell profiles has not been reported. It is well recognized that in addition to the dysregulation of oncogenes or tumor-suppressor genes of the tumor, the progression of cancer is also driven by the interactions between tumor and host cells that stimulate immune and inflammatory responses [26]. The correlation between peripheral inflammatory cells in CRC and both the recurrence and survival of patients has also been revealed in multiple studies [27,28]. RDW is a hematological parameter that indicates the heterogeneity of circulating red blood cells, and high levels of RDW are associated with both systemic inflammation and poor prognosis in CRC patients [29,30,31]. Furthermore, platelet and lymphocyte counts are the indicators for immune response, inflammation, and coagulation status, and were found to be correlated with the prognosis in CRC. An elevated PLR is from the result of an increase in platelet count and/or decrease in lymphocyte count. Elevated platelet counts were reported to be associated with tumor cells with greater metastatic ability [32], through secretion of multiple cellular growth factors such as platelet-derived growth factor, vascular endothelial growth factor, transforming growth factor beta, platelet factor 4, and inflammatory mediators, which are responsible for stimulating the growth and angiogenesis of tumor cells [33,34]. Multiple studies have also revealed that platelets promote both tumor stoma formation and stable tumor–endothelium adhesion, thus enhancing the proliferation and angiogenesis ability of the tumor [35,36]. Hence, by preventing innate immune cells from removing the tumor cells via platelet–tumor cell interaction in the bloodstream, platelets could play a role in supporting metastasis of the cancer [37]. In this study, we showed that tumor miR-187-3p was associated with platelet and PLR levels, but found no correlation with lymphocyte counts. Furthermore, evidence suggests a correlation between an increase in NLR and poor prognosis, such as reductions in overall and disease-specific survival, as well as time to recurrence [38]. Being part of peripheral blood and responsible for the secretion of various growth factors, proteases, and chemokines, neutrophils are considered to potentially support the tumor development and angiogenesis by providing such regulatory proteins [39]. Our study showed that high tumor miR-187-3p levels were associated with high RDW, platelet count, PLR, neutrophil count, and NLR, confirming that its high levels were correlated with poor prognosis. Since these peripheral inflammatory cells can induce tumor progression and recurrence, further investigations are warranted to study the functional effect of miR-187-3p on their levels.

## 5. Conclusions

This study demonstrated that miR-187-3p was significantly repressed in CRC cells. Overexpression of miR-187-3p impaired CRC cell migration and colony formation, and enhanced the response of CRC cells to chemotherapeutic drugs. Clinical data analyses showed that tumor miR-187-3p levels were higher in stage II–IV CRCs when compared to stage I, and lower in distant metastatic tumor cells compared to the primary tumor, suggesting that CRC cells with lower miR-187-3p levels have stronger metastatic ability and can escape from the primary tumor site. High tumor miR-187-3p levels were also associated with poor disease-free and overall survival. Finally, tumor miR-187-3p levels were positively correlates with RDW, PLT, PLR, neutrophil count, and NLR levels, indicating their association with systematic inflammation.

## Figures and Tables

**Figure 1 cells-11-02421-f001:**
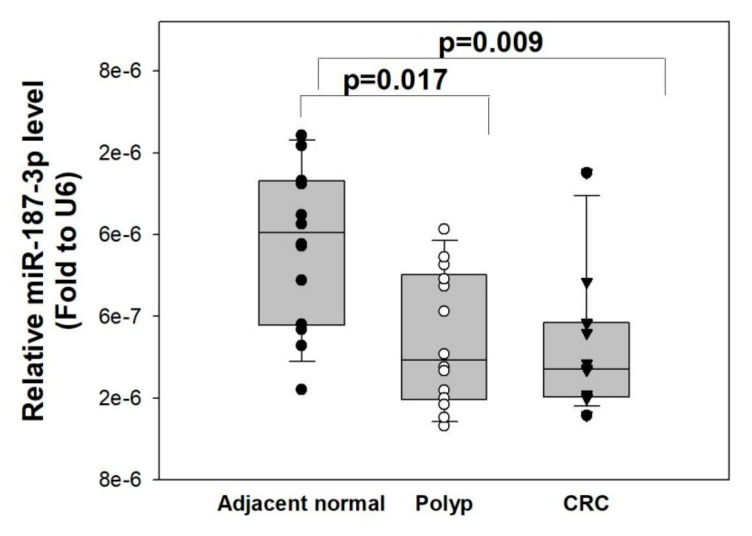
MiR-187-3p was repressed in colorectal polyp and CRC tissues: MiR-187-3p levels were determined in synchronous polyp, CRC, and normal mucosa samples from 14 patients by quantitative RT-PCR. MiR-187-3p was significantly repressed in polyp (*p* = 0.017) and CRC (*p* = 0.009) tissues when compared to adjacent normal tissue. No significant difference was observed between polyp and CRC tissues. The level of miR-187-3p was expressed as 2^(−delta^
^Ct[^^miR-187-3p–^^U^^6])^. Experiments were performed in duplicate, and the data are expressed as the mean ± SEM of three independent experiments.

**Figure 2 cells-11-02421-f002:**
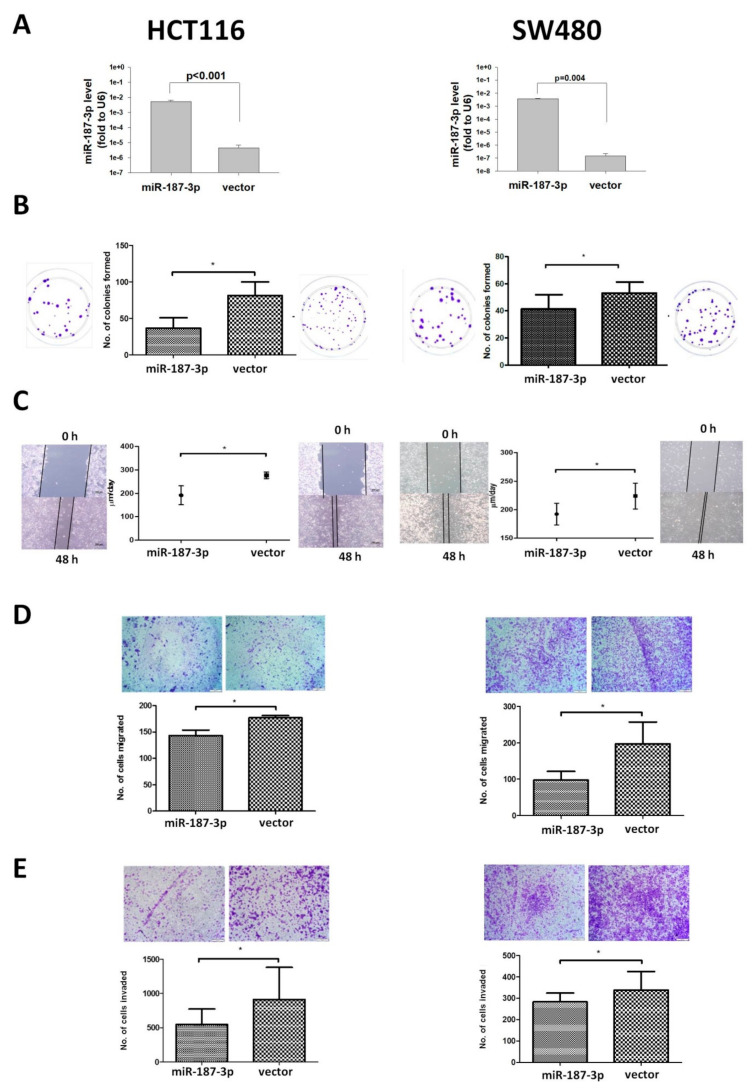
MiR-187-3p overexpression inhibited CRC cell migration and colony formation: (**A**) MiR-187-3p plasmids or control vectors were stably overexpressed in HCT116 or SW480 cells. Quantitative PCR was performed to determine the expression of miR-187-3p in miR-187-3p or control vector clones. (**B**) Colony-formation assay was performed to compare the clonogenic ability of stable miR-187-3p and control vector HCT116 or SW480 cells. (**C**) Wound-healing assay was performed to compare the cell migration rates of stable HCT116 or SW480 cells with miR-187-3p or the control vector 48 h after wound creation. (**D**) Cell migration assay was performed to compare the migration ability of stable HCT116 or SW480 cells with miR-187-3p or the control vector. (**E**) Cell invasion assay was performed to compare the invasion ability of stable HCT116 or SW480 cells with miR-187-3p or the control vector. All experiments were performed in duplicate, and the data are expressed as the mean ± SEM of three independent experiments; * indicates that the difference is statistically significant (*p* < 0.05).

**Figure 3 cells-11-02421-f003:**
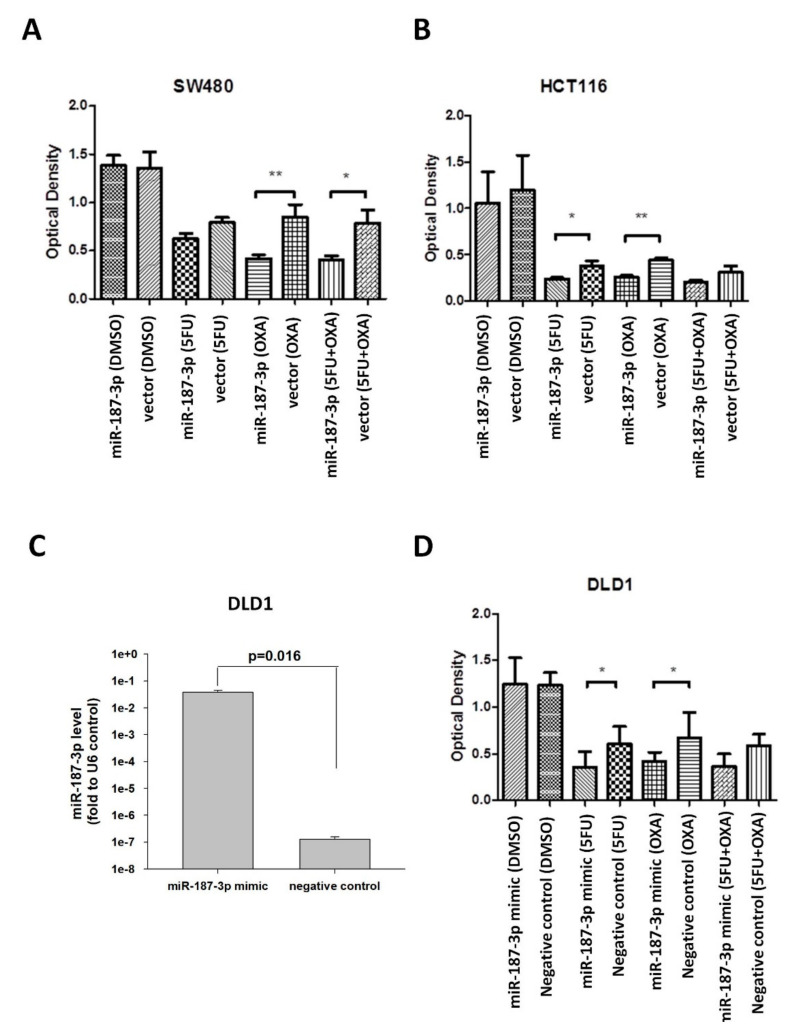
MiR-187-3p overexpression induced response of CRC cells to chemotherapeutic drugs: MTT assay was performed to determine the effect of 72 h treatment with vehicle control (DMSO), 5-fluorouracil (5-FU), oxaliplatin (OXA), or combined 5-FU + OXA on (**A**) stable HCT116-miR-139-3p or control vector cells, (**B**) stable HCT116-miR-139-3p or control vector cells, and (**C**) DLD1 transiently overexpressed miR-187-3p mimic or negative control cells. The left panel shows the miR-187-3p levels in the transfected cells as determined by quantitative PCR. (**D**) The optical density in terms of absorbance at 570 nm was determined after 72 h drug or vehicle treatment. All experiments were performed in triplicate, and the data are expressed as the mean ± SEM of three independent experiments; * and ** indicate that the difference is statistically significant (*p* < 0.05 and *p* < 0.01, respectively).

**Figure 4 cells-11-02421-f004:**
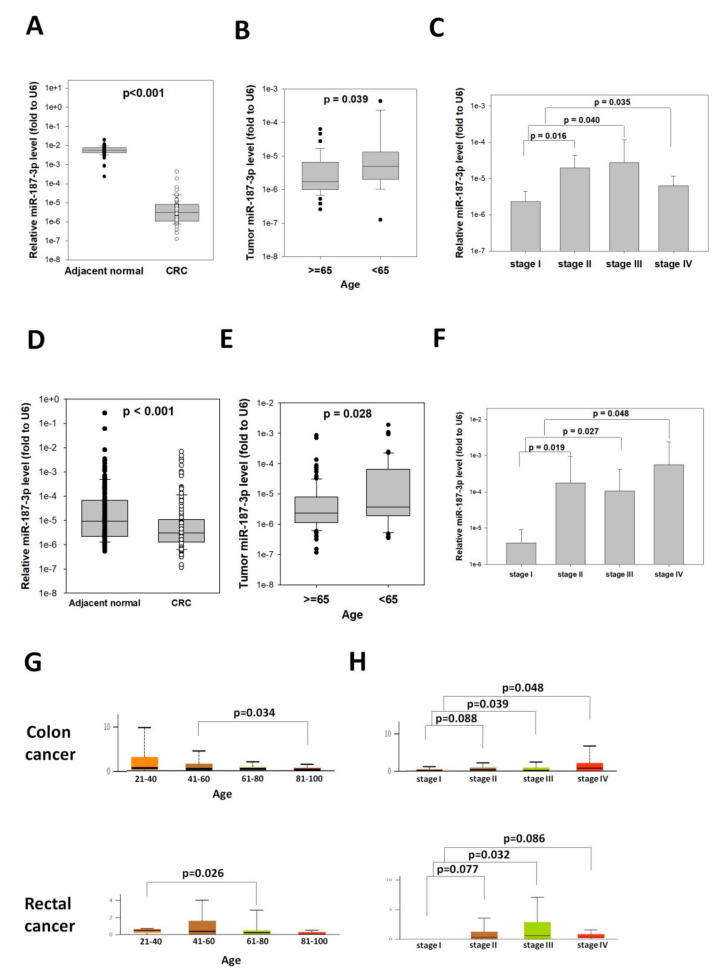
MiR-187-3p levels were associated with the age and tumor stage of CRC patients: (**A**) MiR-187-3p levels in paired CRC and adjacent normal tissues of 56 CRC patients in the study cohort. (**B**) The miR-187-3p levels were significantly lower in patients aged 65 years or above compared to those aged below 65. (**C**) The miR-187-3p levels were significantly higher in stage II–IV CRCs compared to stage I CRCs. (**D**) MiR-187-3p levels in paired CRC and adjacent normal tissues of 160 CRC patients in the validation cohort. (**E**) The miR-187-3p levels were significantly lower in patients aged 65 years or above compared to those aged below 65. (**F**) The miR-187-3p levels were significantly higher in stage II–IV CRCs compared to stage I CRCs. (**G**) TCGA analysis showed that miR-187-3p levels were significantly different between different age groups in patients with colon cancer (upper panel) or rectal cancer (lower panel). (**H**) TCGA analysis showed that miR-187-3p levels were significantly higher or tended to be higher in stage II–IV CRCs compared to stage I CRCs for patients with colon cancer (upper panel) or rectal cancer (lower panel). ∙ and ○ indicated the individual data points of adjacent normal tissues and CRC, respectively.

**Figure 5 cells-11-02421-f005:**
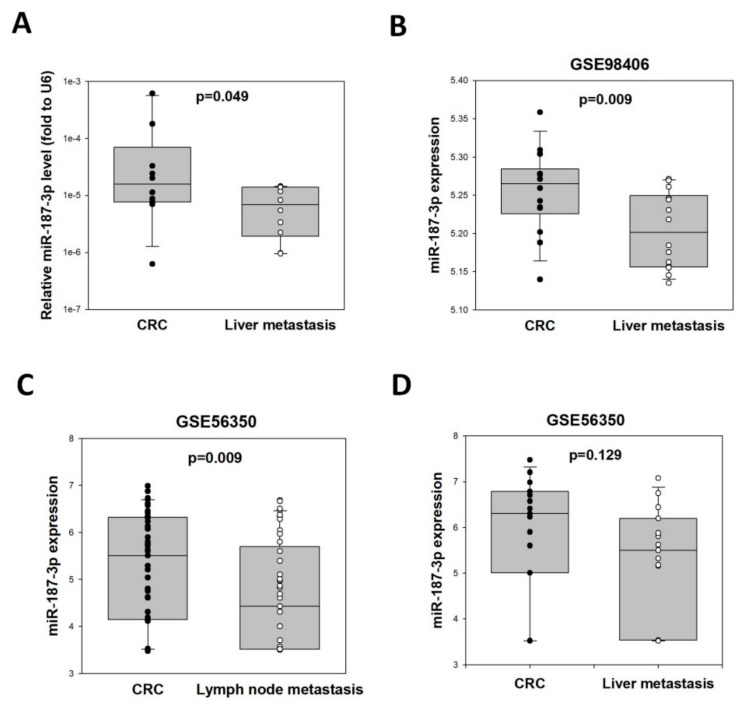
MiR-187-3p levels were lower in metastatic cancer cells compared to primary CRC cells: (**A**) MiR-187-3p levels in paired liver metastasis and primary CRC specimens of 10 patients in this study. (**B**) Data obtained from the GEO dataset GSE98406 showing that miR-187-3p levels were significantly lower in liver metastases comparing to primary CRC cells (*n* = 14). (**C**) Data obtained from the GEO dataset GSE56350 showing that miR-187-3p levels were significantly lower in lymph node metastases compared to primary CRC cells (*n* = 41). (**D**) Data obtained from the GEO dataset GSE56350 showing that there was a trend of lower miR-187-3p levels in paired liver metastases comparing to primary CRC cells (*n* = 14). ∙ and ○ indicated the individual data points of adjacent normal tissues and CRC, respectively.

**Figure 6 cells-11-02421-f006:**
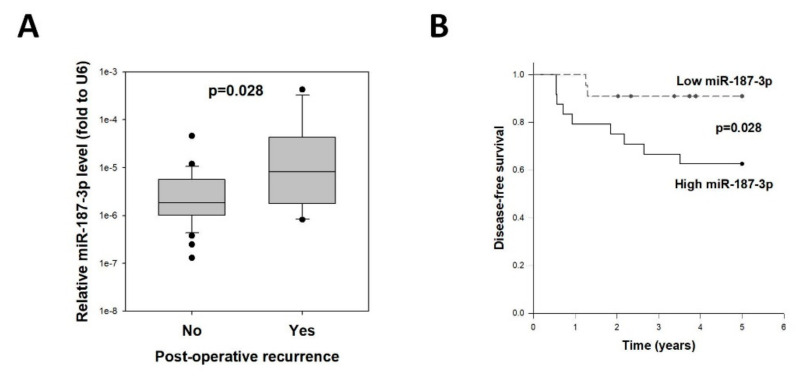
High tumor miR-187-3p levels were correlated with postoperative recurrence: (**A**) Tumor miR-187-3p levels in CRC patients with no recurrence (*n* = 35) or with recurrent disease (*n* = 11) within 5 years following R0 resection for stage I–III CRC. (**B**) Disease-free survival curves for CRC patients with high or low tumor miR-187-3p levels.

**Figure 7 cells-11-02421-f007:**
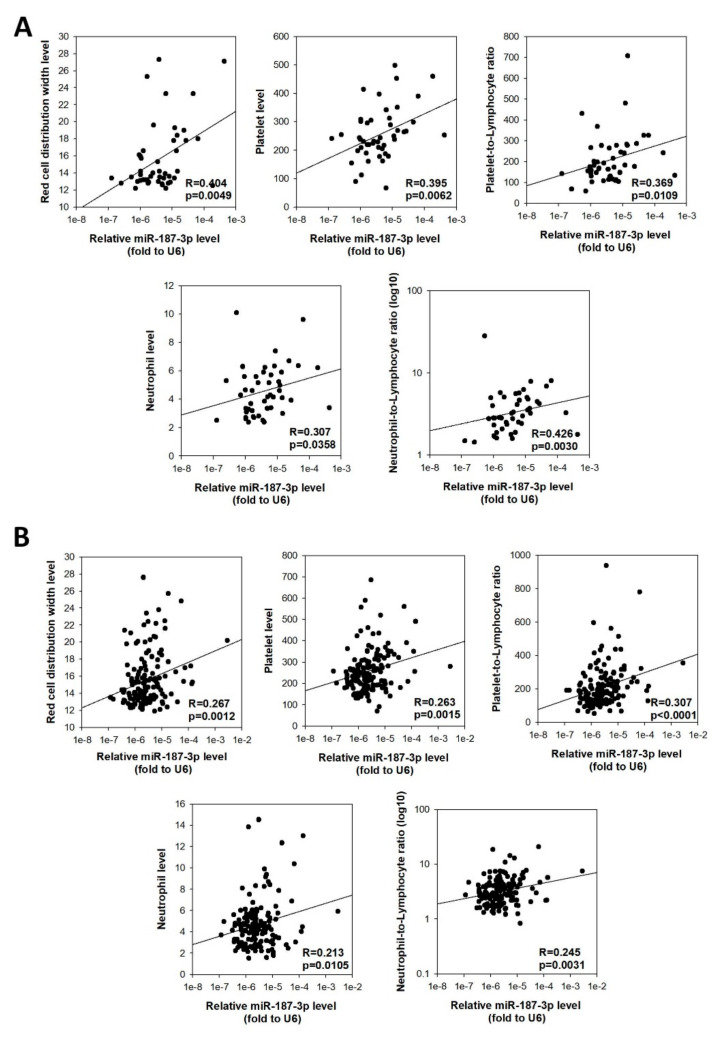
Tumor miR-187-3p levels were correlated with preoperative peripheral inflammation-related blood parameters: Positive correlation was found between tumor miR-187-3p levels and levels of red cell distribution width, platelet count, platelet-to-lymphocyte ratio, neutrophil count, and neutrophil-to-lymphocyte ratio for patients in the (**A**) study cohort (*n* = 54) and (**B**) validation cohort (*n* = 160).

**Figure 8 cells-11-02421-f008:**
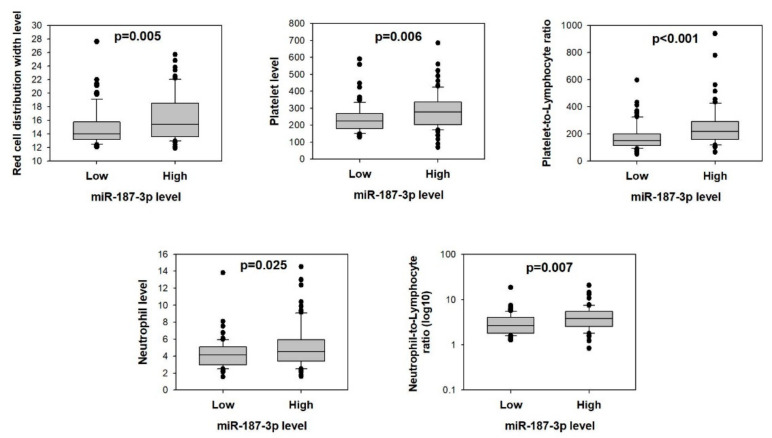
Higher preoperative peripheral inflammation-related blood parameters in CRC patients with higher tumor miR-187-3p levels: Comparison of red cell distribution width, platelet count, platelet-to-lymphocyte ratio, neutrophil count, and neutrophil-to-lymphocyte ratio between CRC patients with high or low tumor miR-187-3p levels in the validation cohort (*n* = 160).

**Figure 9 cells-11-02421-f009:**
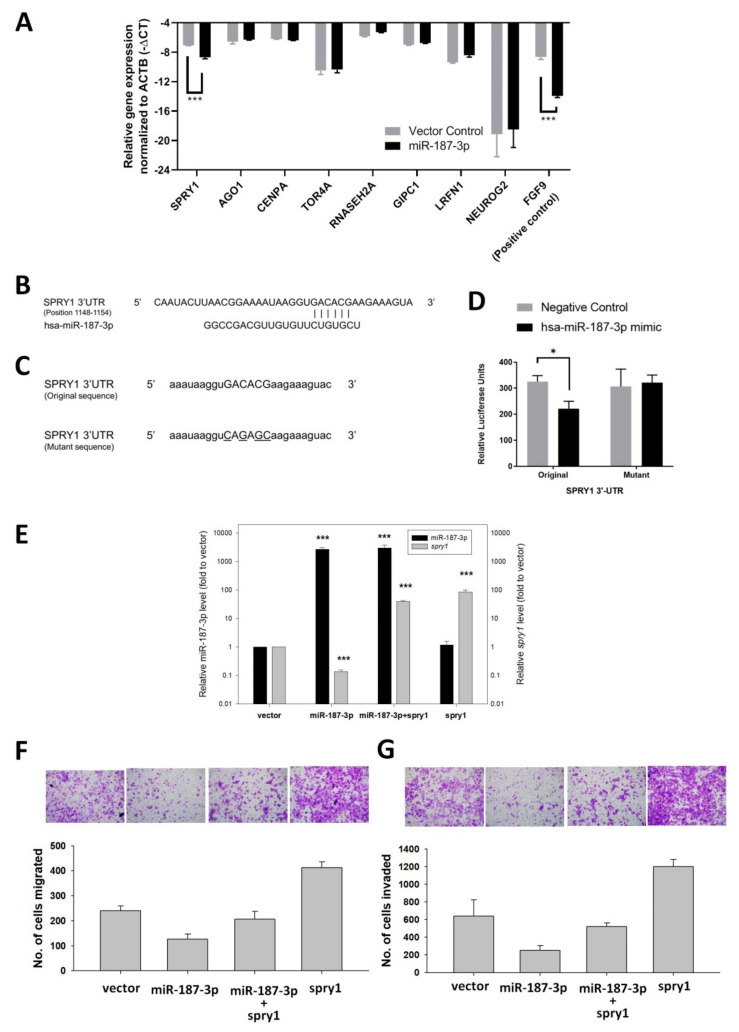
Identification of SPRY1 as a target gene of miR-187-3p in vitro: (**A**) Quantitative RT-PCR-based expression analysis of putative target genes within HCT116 stably transfected with miR-187-3p or the control vector. Expression of target genes was normalized to *ACTB* and reported as –delta Ct (negative delta Ct). FGF9 was used as a positive control. (**B**) Predicted binding site of miR-187-3p within the 3′-UTR region of the *SPRY1* gene. (**C**) Design of SPRY1 (original and mutant) insert sequence within the reporter construct. Uppercase: potential binding site; underline: mutated bases within the potential binding site; lowercase: flanking 3′-UTR bases. (**D**) HCT116 cells were co-transfected with SPRY1 3′-UTR original or mutant sequence reporter constructs with has-miR-187-3p mimic or negative control (empty vector) and Renilla luciferase control vector. The firefly luciferase signal for each transfectant was normalized by its corresponding Renilla luciferase signal and expressed in relative luciferase units. (**E**) Gene expression levels of miR-187-3p and SPRY1 within the HCT116 control vector cells (vector), stable miR-187-3p cells (miR-187-3p)), stable miR-187-3p cells transiently transfected with the SPRY1 expression plasmid (miR-187-3p+SPRY1), or control vector cells transfected with the SPRY1 expression plasmid (SPRY1). (**F**,**G**) Rescue experiment demonstrating the effects of SPRY1 overexpression in HCT116 control vector cells or stable miR-187-3p cells on (**F**) cell migration and (**G**) invasion. Data are expressed as the mean ± SEM of three independent experiments; * and *** indicate that the difference is statistically significant (*p* < 0.05 and *p* < 0.001, respectively).

**Figure 10 cells-11-02421-f010:**
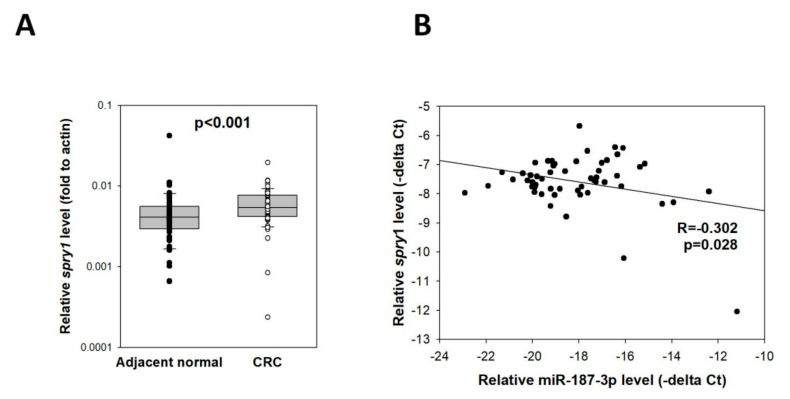
Expression of *SPRY1* and its correlation with miR-187-3p in CRC tissues: (**A**) Quantitative RT-PCR was performed to determine the *SPRY1* levels in 54 paired CRC specimens and adjacent normal mucosa. (**B**) Pearson’s correlation analysis in the CRC cohort showed that there was a significant inverse correlation between miR-187-3p and *SPRY1* (R = −0.302, *p* = 0.028). The levels of miR-187-3p were expressed as –delta Ct (miR-187-3p–U6), whereas levels of *SPRY1* were expressed as –delta Ct (*spry1*-*actin*). ∙ and ○ indicated the individual data points of adjacent normal tissues and CRC, respectively.

## Data Availability

The data presented in this study are available on request from the corresponding author. The data are not publicly available due to privacy. **Acknowlegement**: The authors would like to thank Cheung Him for his technical support.

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
