# Peer review of "High Levels of Tumor miR-187-3p—A Potential Tumor-Suppressor microRNA—Are Correlated with Poor Prognosis in Colorectal Cancer"

_cells, 2022, doi:10.3390/cells11152421_

Round 1
Reviewer 1 Report
In this manuscript, Lui Ng et al showed that potential tumor suppressor microRNA-187-3P correlates with poor prognosis in colorectal cancer cells. These findings are potentially interesting. The manuscript could be further strengthened with a few additional experiments denoted below.
1. Please replace the data in Figure 1C because of difference in brightness, it is not easy to distinguish between Control (Vector) and miR-187-3p group.
2. In figure 3, is there an experimental result for the target gene of the microRNA-187-3p in the result (By western blotting or qRT-PCR)?
3. In figure 5, is there an experimental result for the target genes which are related for metastasis?
4. Most of the data only showed high and low expression, and information on specific target genes is insufficient.
Author Response
Response to Reviewer 1 Comments
In this manuscript, Lui Ng et al showed that potential tumor suppressor microRNA-187-3P correlates with poor prognosis in colorectal cancer cells. These findings are potentially interesting. The manuscript could be further strengthened with a few additional experiments denoted below.
Thank you very much for the critical comments which significantly improve the completeness of our study.
Point 1: Please replace the data in Figure 1C because of difference in brightness, it is not easy to distinguish between Control (Vector) and miR-187-3p group.
Response 1: Figure 2C is replaced with another representative photo with better brightness.
Point 2 In figure 3, is there an experimental result for the target gene of the microRNA-187-3p in the result (By western blotting or qRT-PCR)?
Response 2: We have included the qRT-PCR result of a novel target gene of miR-187-3p, spry1, in the revised manuscript Figure 9.
Point 3: In figure 5, is there an experimental result for the target genes which are related for metastasis?
Response 3: We have included the experiment of identifying a novel target gene of miR-187-3p, spry1, in the revised manuscript and performed rescue experiment to demonstrate the role of spry1 in regulating miR-187-3p-regulated metastasis (Figure 9).
Point 4: Most of the data only showed high and low expression, and information on specific target genes is insufficient.
Response 4: In the revised manuscript, we have included the experiment of identifying spry1 as a novel target gene, investigated the role of spry1 in regulating miR-187-3p-regulated metastasis, and confirmed their negative correlation in CRC specimens (Figure 9 and 10). The significance of spry1 is also discussed in the discussion section.
Reviewer 2 Report
In this manuscript, Ng et al. reported the function of miR-187-3p as a tumor suppressor miRNA in colorectal cancer (CRC). This work is straightforward in terms of designing the experiment to fully capture the miRNA changes between polyp/CRC/normal tissues, carrying out overexpression experiments to validate the functions, and investigating the role of miR-187-3p in metastasis. The authors discovered that miR-187-3p remains low expression in the tumor, and overexpression of it will impair the tumor cells’ migration and invasion. This work would be suitable for this journal; however, I have found a few issues:
1, It’s not clear at all to use lines to represent the expression values, in Figure 1, Figure 4 A and D, and Figure 5 A-D. These lines won’t show the distribution of the data, and most of them overlapped. I suggest they use boxplots with original data overlayed as dots.
2, In 3.7, the authors mentioned that although miR-187-3p remained low expression in the tumors, its level will increase when the tumor progresses. This part needs more clarification. The authors state that miR-187-3p may serve as a biomarker for recurrence, when will be the best time to examine the miR-187-3p level, if 5 years is too late? Will miR-187-3p increase gradually or after the new tumor reaches a certain stage? This seems to contradict the finding that miR-187-3p remains low expression in the tumor.
Overall, it’s promising to see miR-187-3p shows a strong and specific correlation with CRC, and I suggest that the authors add more future directions to understand the mechanisms behind this, such as the target of miR-187-3p and potential therapeutic strategies.
Author Response
Response to Reviewer 2 Comments
In this manuscript, Ng et al. reported the function of miR-187-3p as a tumor suppressor miRNA in colorectal cancer (CRC). This work is straightforward in terms of designing the experiment to fully capture the miRNA changes between polyp/CRC/normal tissues, carrying out overexpression experiments to validate the functions, and investigating the role of miR-187-3p in metastasis. The authors discovered that miR-187-3p remains low expression in the tumor, and overexpression of it will impair the tumor cells’ migration and invasion. This work would be suitable for this journal; however, I have found a few issues:
Thank you very much for the comprehensive comments from this reviewer.
Point 1: It’s not clear at all to use lines to represent the expression values, in Figure 1, Figure 4 A and D, and Figure 5 A-D. These lines won’t show the distribution of the data, and most of them overlapped. I suggest they use boxplots with original data overlayed as dots.
Response 1: These figures are replaced by boxplots with original data overlayed as dots in the revised manuscript.
Point 2: In 3.7, the authors mentioned that although miR-187-3p remained low expression in the tumors, its level will increase when the tumor progresses. This part needs more clarification. The authors state that miR-187-3p may serve as a biomarker for recurrence, when will be the best time to examine the miR-187-3p level, if 5 years is too late? Will miR-187-3p increase gradually or after the new tumor reaches a certain stage? This seems to contradict the finding that miR-187-3p remains low expression in the tumor.
Response 2: We understand that this is quite surprising that although miR-187-3p remained low expression in the tumors, its level will increase when the tumor progresses and its high level associated with poor prognosis. But as mentioned in the discussion section, we found that this unusual and contradictory observation has also been reported for another miRNA, miR-485-5p, which was repressed in CRC but its high expression was associated with poor prognosis, which is also able to impair cell invasion and migration which is similar to the functional effect of miR-187-3p in our study. Moreover, miR-187-3p is also repressed in HCC and functionally inhibits metastasis of HCC both in vitro and in vivo. On the other hand high miR-187-3p level in HCC is also associated with poor prognosis. So these literatures could be a support to our finding that the association between poor prognosis and high expression of these tumor suppressor miRNAs is a result of progression of tumor and subsequent dissemination of metastatic tumor cell, which expressed low level of miR-187-3p or miR-485-5p, from the primary tumor site.
We added the following content in the discussion section:
“Our clinical experiment showed that miR-187-3p level was higher in stage II, III and IV CRCs when compared to stage I CRC, albeit its level was significantly lower than the paired non-tumor tissue (data not shown), it is rational to postulate that the association between poor prognosis and high expression of these tumor suppressor miRNAs is a result of progression of tumor and subsequent dissemination of metastatic tumor cell, which expressed low level of miR-187-3p or miR-485-5p, from the primary tumor site. Our subsequent experiment which showed that miR-187-3p level was significantly lower in liver metastasis specimen comparing to paired CRC supported this postulation.”
Regarding the potential of applying miR-187-3p as a biomarker for recurrence, we compared the miR-187-3p level in tumors collected from two groups of CRC patients (1) patients who developed recurrent disease within 5 years after operation; and (2) patients remained disease-free in 5 years. This study showed that CRC specimens with a higher miR-187-3p has a higher risk for developing recurrence within 5 years post-operation. Hence we suggested that the tumor miR-187-3p level is a potential predictive biomarker for recurrence. We added the following paragraph in the revised manuscript:
“This study also suggested that tumor miR-187-3p level was a predictive biomarker for CRC recurrence. Tumor miR-187-3p level was significantly higher in stage I to III CRC patients who developed recurrent disease within 5 years post-operation, when compared to those showed no recurrence. Moreover, CRC patients with higher tumor miR-187-3p expression had a significantly worse disease-free survival when compared to those with low tumor miR-187-3p level. Further investigation in another cohort of patients is warranted to validate our finding. We hope that by identifying a panel of promising biomarker for predicting risk of recurrence, such as miR-187-3p in this study, better post-operative treatment can be designed for CRC patients.”
Point 3: Overall, it’s promising to see miR-187-3p shows a strong and specific correlation with CRC, and I suggest that the authors add more future directions to understand the mechanisms behind this, such as the target of miR-187-3p and potential therapeutic strategies.
Response 3: In the revised manuscript, we have included the experiment of identifying spry1 as a novel target gene, investigated the role of spry1 in regulating miR-187-3p-regulated metastasis, and confirmed their negative correlation in CRC specimens (Figure 9 and 10). The significance of spry1 is also discussed in the discussion section.
Round 2
Reviewer 2 Report
The revised manuscript can be accepted.